# A Monoallelic Variant in *REST* Is Associated with Non-Syndromic Autosomal Dominant Hearing Impairment in a South African Family

**DOI:** 10.3390/genes12111765

**Published:** 2021-11-06

**Authors:** Noluthando Manyisa, Isabelle Schrauwen, Leonardo Alves de Souza Rios, Shaheen Mowla, Cedrik Tekendo-Ngongang, Kalinka Popel, Kevin Esoh, Thashi Bharadwaj, Liz M. Nouel-Saied, Anushree Acharya, Abdul Nasir, Edmond Wonkam-Tingang, Carmen de Kock, Collet Dandara, Suzanne M. Leal, Ambroise Wonkam

**Affiliations:** 1Department of Medicine, Division of Human Genetics, Faculty of Health Sciences, University of Cape Town, Cape Town 7925, South Africa; MNYNOL006@myuct.ac.za (N.M.); Kalinka.popel@uct.ac.za (K.P.); esohkevin4@gmail.com (K.E.); wonkamedmond@yahoo.fr (E.W.-T.); carmen.dekock@uct.ac.za (C.d.K.); collet.dandara@uct.ac.za (C.D.); 2Center for Statistical Genetics, Sergievsky Center and the Department of Neurology, Columbia University Medical Center, New York, NY 10032, USA; is2632@cumc.columbia.edu (I.S.); tb2890@cumc.columbia.edu (T.B.); lmn2152@cumc.columbia.edu (L.M.N.-S.); aa4471@cumc.columbia.edu (A.A.); sml3@cumc.columbia.edu (S.M.L.); 3Department of Pathology, Division of Haematology, Faculty of Health Sciences, University of Cape Town, Observatory, Cape Town 7925, South Africa; leonardo.rios@uct.ac.za (L.A.d.S.R.); Shaheen.mowla@uct.ac.za (S.M.); 4Medical Genetics Branch, National Human Genome Research Institute, National Institutes of Health, Bethesda, MD 20892, USA; cedrik.ngongang@nih.gov; 5Synthetic Protein Engineering Lab (SPEL), Department of Molecular Science and Technology, Ajou University, Suwon 443-749, Korea; anasirqau@gmail.com; 6Taub Institute for Alzheimer’s Disease and the Aging Brain, Columbia University Medical Center, New York, NY 10032, USA

**Keywords:** *REST*, RE1-silencing transcription factor, non-syndromic hearing impairment, South Africa, Africa, *DFNA27*

## Abstract

Hearing impairment (HI) is a sensory disorder with a prevalence of 0.0055 live births in South Africa. DNA samples from a South African family presenting with progressive, autosomal dominant non-syndromic HI were subjected to whole-exome sequencing, and a novel monoallelic variant in *REST* [c.1244GC; p.(C415S)], was identified as the putative causative variant. The co-segregation of the variant was confirmed with Sanger Sequencing. The variant is absent from databases, 103 healthy South African controls, and 52 South African probands with isolated HI. In silico analysis indicates that the p.C415S variant in *REST* substitutes a conserved cysteine and results in changes to the surrounding secondary structure and the disulphide bonds, culminating in alteration of the tertiary structure of REST. Localization studies using ectopically expressed GFP-tagged Wild type (WT) and mutant *REST* in HEK-293 cells show that WT REST localizes exclusively to the nucleus; however, the mutant protein localizes throughout the cell. Additionally, mutant REST has an impaired ability to repress its known target *AF1q*. The data demonstrates that the identified mutation compromises the function of *REST* and support its implication in HI. This study is the second report, worldwide, to implicate *REST* in HI and suggests that it should be included in diagnostic HI panels.

## 1. Introduction

Hearing impairment (HI) is a sensory disorder that affects 466 million people [1]. HI affects approximately 1 in 1000 newborns worldwide, but it has an estimated prevalence of 5.5 in 1000 live births in South Africa [2]. HI is defined as the inability to detect sound better than 25 dB in the better hearing ear and it is considered disabling HI when a child cannot hear better than 30 dB and an adult cannot hear better than 40 dB in their better hearing ears [1].

Hearing impairment has a heterogeneous etiology with the causes of HI being broadly classified as either genetic, environmental, or due to unknown factors. Genetic factors may account for 50% of HI in high income countries [3]. Approximately 121 genes have been identified as being associated with non-syndromic hearing impairment (NSHI) [4]. Non-syndromic hearing impairment accounts for about 70% of HI cases of genetic origin and is inherited on an autosomal recessive (AR) mode in approximately 80% of cases, while autosomal dominant (AD) account for 15 to 20% [5]. Variants in *GJB2* and *GJB6* genes are the major contributors to NSHI in Europeans, Asians, and Arabs [6,7,8,9,10], but are infrequent in most populations of African descent, including black South Africans. In addition, using data from the genome aggregation database (gnomAD) database, for ARNSHI the prevalence of identified likely pathogenic and pathogenic (PLP) variants [11] was estimated at 96.9 per 100,000 individuals for Ashkenazi Jews for ARNSHI based on sequence data compared to only 5.2 per 100,000 individuals among Africans/African Americans [12] indicating for African populations many variants remain to be discovered. Therefore, there is an urgent need to investigate HI in African populations using next generation sequencing and multiplex HI families.

RE1-silencing transcription factor (*REST)* is a transcriptional repressor that binds to a 23-bp RE1 consensus sequence in the promoter region of its target genes [13,14,15]. The *DFNA27* locus, containing *REST*, was mapped in 2008 in a North American family [16]. *REST* was reported as a strong candidate associated with HI in the family in 2018, wherein an intronic variant (NC_000004.12:g.56927594CG) was identified [17]. *REST* contains 4 exons and encodes a 1097aa protein that restricts the expression of neuronal genes in non-neuronal cells [13,14,18]. This variant (g.56927594CG) results in the prevention of alternative splicing of the *REST* mRNA which is necessary for the regulation of REST in the inner ear [17].

In this study, whole exome sequence (WES) data was generated from DNA samples obtained from a non-consanguineous South African family. The family presented with progressive ADNSHI and a novel monoallelic variant in *REST* (NM_005612.5:c.1244GC, p.C415S, GRCh37:4:57796268:G:C), in the locus DFNA27, was identified.

## 2. Materials and Methods

### 2.1. Participants’ Recruitment

The participants’ selection process followed the protocol reported by reported Bosch et al. (2014a, 2014b) [19,20]. The hearing-impaired members of a South African Xhosa family of African ancestry (Family 1, Figure 1a) were identified through a national recruitment program for the deaf. The detailed personal history and medical records of the hearing-impaired participants (mother and son) were reviewed by two medical geneticists (C.T.N and A.W.). A general systemic and otological examination was performed, including pure tone audiometry. We followed the recommendation number 02/1 of the Bureau International d’Audiophonologie (BIAP), Belgium.

In addition, a total of 52 unrelated probands with sporadic NSHI of putative genetic origin comprising of both Black South African and South Africans of Mixed Ancestry (Appendix A) were recruited, to investigate the frequencies of possible PLP variants identified. Moreover, 103 healthy Black South African controls without personal or familial history of HI were randomly recruited at outpatient clinics at Groote Schuur Hospital, Cape Town, South Africa (Appendix A).

### 2.2. Molecular Methods

#### 2.2.1. DNA Extraction

Peripheral Blood samples were obtained from the proband (IV.3) and three family members: the proband’s affected mother (III.5), unaffected half-brother (IV.4), and maternal grandmother (II.4) (Figure 1a). DNA was extracted using the chemagen 360 Instrument (PerkinElmer, Massachusetts, USA) per the manufacturer’s instructions.

#### 2.2.2. Whole Exome Sequencing

DNA samples, from the two affected members of Family 1, underwent whole-exome sequencing at OmegaBioservices (Norcross, GA, USA). An Illumina Nextera Rapid Capture Kit^®^ (37Mb; San Diego, CA, USA) was used for library preparation, according to the manufacturer’s instructions. The library was then sequenced using the Illumina HiSeq 2500 sequencer using the pair-end 150bp run format. The sequencing data were processed using the Illumina DRAGEN Germline Pipeline v3.2.8.

### 2.3. Bioinformatics Analysis

#### 2.3.1. Alignment and Quality Checking

High-quality reads were aligned to the human GRCh37/hg19 human reference genome using the DRAGEN software (version 05.021.408.3.4.12). Variants were called following sorting and marking of duplicates and individual genomic variant call files (gVCF) were generated. Joint variant calling for single nucleotide variations (SNV) and Insertion/Deletions(Indels) was performed using the genome analysis toolkit (GATK) software (version 4.0.6.0) [21]. For those individuals that underwent exome sequencing, their sex of was verified using plink (version 1.9) [22,23]. Additionally, identity-by-descent sharing in plink (version1.9) and Kinship-based Inference for GWAS (KING) algorithms were used to verify their familial relationships [22,23,24].

#### 2.3.2. Variant Annotation and Filtering

ANNOVAR and custom scripts were used for variant annotation and filtering [25]. Variants were initially prioritized using an AD mode of inheritance. Rare variants for all populations in the genome aggregation database (gnomAD) [11,26], with minor allele frequencies 0.0005, and known likely pathogenic and pathogenic variants in ClinVar were retained [26].

Functional prediction of missense variants was performed by annotating dbNSFP (version4.0) [27,28]. dbNSFP includes Sorting Intolerant from Tolerant (SIFT), polymorphism phenotyping v2 (PolyPhen-2) × 2, MutationAssessor, the likelihood ratio test (LRT), Mendelian clinically applicable pathogenicity (M-CAP) score, Rare Exome Variant Ensemble Learner (REVEL), MutPred, PROtein Variation Effect Analyzer (PROVEAN), MetaSVM, and MetaLR [27,28]. Whereas the tools MutationTaster, Eigen, Eigen-PC, functional analysis through Hidden Markov models (FATHMM-MKL), combined annotation dependent depletion (CADD) score, and deleterious annotation of genetic variants using neural networks (DANN) were used to evaluate both coding and non-coding variants.

Splice site variants were analyzed using dbscSNV which used Adaptive boosting and random forest scores [29]. This tool allowed for the analysis of the deleterious effect of variants within conserved splicing regions, −3 to +8 at the 5′ splice site and −12 to +2 at the 3′ splice site [29]. Furthermore, the conservation of nucleotides and amino acids was estimated, at which the variations occur, using phyloP, Genomic Evolutionary Rate Profiling (GERP), SiPhy, and phastCons scores [27,28,30,31].

The online Mendelian inheritance in man (OMIM) [32,33], ClinVar, and gnomAD were used to determine if there were known associations between identified genes and/or variants and HI. Variants were considered to be putatively causative if they occurred in known HI genes or genes that were expressed in the inner ear, if the variant had a predicted effect on protein function or the mRNA, and if the variation segregated with HI within the family.

### 2.4. Direct Cycle Sequencing

Direct cycle sequencing was performed to validate the segregation of the candidate variation in the family. Cycle sequencing was also performed on 52 probands with sporadic NSHI of putative genetic origin, and 103 presumably healthy, ethnically controls who had no family history of HI. Primers (forward 5′-GTTCTTTAGTAGTGCTTGAGG-3′ and reverse 5′-GGTGACTACCAGAACTCG-3′) that target the variant of interest in exon 4 were designed based on the genomic sequence of *REST* (OMIM: 600571), from Ensembl [34]. The primer specificity was evaluated using primerBlast [35] and In silico PCR. The annealing temperature was determined to be 60 °C for 45 s and fragment elongation occurred at 72 °C for 1 min. Sequencing of the PCR was performed using BigDye™ Terminator v3.1 Cycle Sequencing Kit and an ABI 3130XL Genetic Analyzer^®^ (Applied Biosystems, Foster City, CA, USA). UniPro UGENE (version 38.1) [36].

### 2.5. Secondary Structure Analysis and Multiple Sequence Alignment

The secondary structure of the wild-type (WT) and mutant (MT) REST was viewed using psipred 4.0 [37,38] and this was followed by performing a multiple sequence alignment (MSA). A PSI-Blast search was performed for REST, using the non-redundant protein data and default search parameters. The PSI-Blast hits were retrieved as FASTA files and the MSA was performed using CLUSTAL Omega (version 1.2.4) [39] and the MSA was viewed using Jalview (version 2.11.1.4) [40].

### 2.6. Protein Modelling and Disulphide Bond Formation

The three-dimensional structure of the longest isoform of REST was used to generate protein models for the WT and MT REST. A homology model of the WT and MT REST was constructed using MODELLER (version 9.4) [41,42,43,44], based on the available crystal structure (6DU2 [45]) as a template. PYMOL Viewer (version 2.4) [[46],] was used for visualization of the structure and image processing. The disulphide bonds within the tertiary structure of REST were analyzed using DiANNA 1.1 Web Server [47,48,49]. Finally, the domains of REST were generated using InterPro [50].

### 2.7. Localisation and Expression Analysis

Mammalian expression plasmids expressing GFP-tagged REST (NM_001193508), and myc-DDK-tagged REST (RG235166 and RC235166) were purchased from Origene (Rockville, MD, USA), and used as templates for generating mutant versions using custom primers (Forward primer 5′ CTTCAAATCTAAGCATCCTACTT**C**TCCTAATAAAACAATGGATGTC 3′ and reverse primer 5′ GACATCCATTGTTTTATTAGGA**G**AAGTAGGATGCTTAGATTTGAAG 3′) acquired from Whitehead Scientific (Stikland, Western Cape, South Africa), based on the protocol adapted from Stratagene QuikChange system. Site-directed mutagenesis (SDM) was performed using the KAPA HiFi HotStart ReadyMIx (Roche, Basel, Switzerland). The plasmids were sequenced by Inqaba Biotec (Gauteng, South Africa) to determine if the SDM was successful.

#### 2.7.1. Cell Culture, Transfections and Visualization Using Confocal Microscopy

HEK-293 human embryonic kidney cells [51] were cultured in Dulbecco’s Modified Eagle Medium (DMEM) (Thermo Fisher Scientific, Waltham, MA, USA) supplemented with 10% (*v*/*v*) fetal bovine serum (Thermo Fisher Scientific, Waltham, MA, USA) and 1% (*v*/*v*) penicillin/streptomycin (Sigma-Aldrich, St. Louis, MO, USA). Cells were cultured in a humidified incubator at 37 °C with 5% CO_2_. Mycoplasma contamination was screened using bisBenzimide H 33342 trihydrochloride (14533, Sigma-Aldrich, St. Louis, MO, USA) DNA staining. The HEK-293 cells were plated in 35 mm dishes (density 4 × 10^4^ cells per mL) 16 h before transfection. Cells were transiently transfected using X-tremeGENE™ HP DNA Transfection Reagent according to the manufacturer’s instructions (Roche, Basel, Switzerland), with 250 ng of plasmid (Empty, GFP-only, GFP-tagged WT REST or GFP-tagged MT REST). Live viewing was performed 24 h after transfection, using a Zeiss LSM8800 with Airyscan confocal microscope (Zeiss, Oberkochen, Germany). Cells were spiked with 1 in 100,000 Hoechst for co-visualization of nuclear material. The detector of the confocal was the photomultiplier tube (PMT) and allowed detection of the green fluorescence signal through the Argon laser at 488 nm. Images were visualized and processed using the ZEN Blue Software (latest version) provided by Zeiss (Zeiss, Oberkochen, Germany).

#### 2.7.2. Quantitative Real-Time PCR

HEK-293 cells were plated (density 8 × 10^4^ cells per ml) in 12-well plates, 16 h before transfection. Cells were transfected using X-tremeGENE™ HP DNA Transfection Reagent, according to the manufacturer’s instructions, with 500 ng of either the WT or MT *REST* plasmid containing the myc-DDK tag and non-transfected cells received X-tremeGENE™ HP DNA Transfection Reagent (Roche, Basel, Switzerland) with no plasmid DNA. Total RNA was extracted 24 h post-transfection using the High Pure RNA Isolation Kit (Roche, Basel, Switzerland) per the manufacturer’s instructions, quantified with Nanodrop1000 (Thermo Fisher Scientific, Waltham, MA, USA), and verified using agarose gel electrophoresis. cDNA was produced from 500 ng of total RNA using the iScript™ Reverse Transcription Supermix for RT-qPCR (BioRad, Hercules, CA, USA), according to the manufacturer’s instructions, and used in qPCR experiment using *AF1q* primers (5′ GGACCCTGTGAGTAGCCAGT 3′, for the forward primer, and 5′ TTGCCAACGCTGCTGTCTTT 3′, for the reverse primer) and *GAPDH* as an internal control (5′ GAAGGCTGGGGCTCATTT 3′, for the forward primer, and 5′ CAGGAGGCATTGCTGATGAT 3′, for the reverse primer). qPCR was performed using KAPA SYBR^®^ FAST (Roche, Basel, Switzerland) on the Rotor-Gene Q (Qiagen, Hilden, Germany) real-time PCR machine. The comparison was made between the negative control, WT and MT using the fold-change (2−ΔΔCT), where the control group was set to 1.

The expression data, from two experiments, was pooled and the mean and standard deviation was determined. The significance of the data was evaluated using the Student’s *t*-test.

#### 2.7.3. Luciferase Assay

HEK-293 cells were plated (density of 8 × 10^4^ cells per ml) in 12-well plates, 16 h before transfection. Cells were transfected using X-tremeGENE™ HP DNA Transfection Reagent (Roche, Basel, Switzerland), according to the manufacturer’s instructions with 400 ng of either WT or MT *REST* plasmid containing the myc-DDK tag or empty pCMV plasmid and 200 ng of pGL2-basic containing *AF1q*. The cells were harvested and 36 h and lysed using the Passive Lysis Buffer (Promega, Madison, WI, USA). The lysate was frozen at −80 °C overnight before thawing and centrifugation at 12,000 rpm at 4 °C. The luciferase assay was performed according to the manufacturer’s instructions and luminometer readings were recorded from the GloMax^®^-Multi Detection System (Promega, Madison, WI, USA).

The expression data, from two experiments, was pooled and the mean and standard deviation was determined. The significance of the data was evaluated using the Student’s *t*-test.

## 3. Results

### 3.1. Phenotypic Description

Despite the small size of the family, the most likely mode of inheritance was ADNSHI. From anamnesis, we did no identified any environmental factors as a possible cause of HI, and no HI participant had a history of ophthalmological clinical expression (blurred or distorted vision, photophobia, eye pain, etc.), or any neurological symptoms such as vertigo or dizziness. Additionally, no vestibular, neurologic, or any other systemic abnormalities were detected by physical examination.

A medical history of prelingual progressive HI was described for the two affected family members; however, prior to this study, no formal audiological assessment was performed on the affected mother and the unaffected grandmother and unaffected half-brother. Audiological assessment of the proband and his mother revealed symmetrical bilateral sensorineural HI (Figure 1b).

The index patient (IV.3) was 12 years old at the time of the recruitment. He was diagnosed with HI at three years of age. He has progressive HI, air conduction thresholds had decreased in his 2019 audiogram when compared to his 2017 audiogram, according to the report from his audiologist. He has severe (pure tone average was evaluated as 71 dB at 500, 1000 and 2000 Hz), sensorineural HI in both ears. He previously had surgery on his ears, to insert grommets and had undergone speech therapy and was using hearing aids when first contacted. There were no associated anomalies in the patient and the parents were unrelated. The HI was determined to be familial non-syndromic HI and putatively of autosomal dominant inheritance.

The affected mother (III.5); was 37 years old at the time of recruitment. She presented with severe HI, in both ears (pure tone average was evaluated as 65 dB in right ear and 67 dB in left ear, at 500, 1000 and 2000 Hz), that was identified and diagnosed at 27 years of age. Her HI is sensorineural in both ears. She had not undergone any speech therapy but used hearing aids when first contacted. She did not present with any associated anomalies and her parents were unrelated.

Conventional pure tone audiometry indicated a mild high frequency hearing impairment in the half-brother of the proband (IV.4). A lack of focus was, however, noted in the child during the assessment and is the most likely cause of the result, rather than the child having high frequency HI.

### 3.2. WES Identification of Candidate Novel Variant in REST

The average target region coverage was about 225×, with 96.30% of the target region being covered to a depth of 10× or more. Through using the filtering criteria described in the methods section, a candidate variant was identified in a known candidate HI gene *REST* (OMIM: 600571). The variation was present in the proband and mother, and thus segregated with the HI phenotype. The variant was confirmed in the mother and the proband through direct Sanger sequencing, and absent in the unaffected younger half-brother and the putatively unaffected grandmother. The novel NM_005612.4:c.1244GC; p.(C415S) variant occurs in the 4th exon and within the lysine-rich protein domain (Appendix A) and was predicted to be damaging by 16 of the 17 bioinformatics tools used (Appendix A), including MutationTaster, FATHMM-MKL, Eigen-PC, CADD, and DANN. The variant was predicted to occur in a conserved position of the genome and was absent from gnomAD, UK10K, Greater Middle East (GME) variome project databases, ClinVar as well as the Single Nucleotide Polymorphism Database (dbSNP). Based on the hearing loss specific American College of Medical Genetics and Genomics (ACMG) guidelines for the interpretation of sequence variants, the variant [52], was classified as likely pathogenic.

Additional heterozygous variants were identified in the proband in two known syndromic autosomal recessive HI genes. This includes a variant of unknown significance (VUS) in *CDH23* (c.5653CT), previously reported as a VUS in a case with Usher Syndrome, [53] and a ClinVar reported pathogenic variant in *NDUFAF3* (c.188dupA) [26], a gene associated with Mitochondrial complex I deficiency, nuclear type 18. These two variants are displayed Appendix A and were found in a heterozygous state in the proband and are both absent in the mother. Thus, they do not segregate with HI in the family and the proband is merely a carrier of these variants.

### 3.3. Sanger Sequencing Confirmation of the Variant

Sanger sequencing confirms the monoallelic candidate variants and its co-segregation with the HI phenotype (Figure 1a,c). The two affected individuals (III.5, and IV.3) were heterozygous for the variant. The unaffected maternal grandmother (II.4), and an unaffected bother (IV.4) did not have the variant (Figure 1a,c).

This variant was not detected in the 103 controls or 52 sporadic NSHI South African probands of Black or Mixed Ancestry (Appendix A). The demographic information of the controls is presented in Appendix A.

### 3.4. Analysis of the REST p.(C415S) Variant on the Protein

#### 3.4.1. Evolutionary Conservation of Amino Acids

Multiple sequence alignment of *REST* from human and other species retrieved from the non-redundant database using PSI-BLAST indicates loss of a highly conserved cysteine residue at position 415 in the protein sequence (Figure 2a). This was consistent with the GERP++RS score of 4.96 (scores of 4–6 indicate fewer observed substitutions as compared to what is expected under neutrality) indicating a strong evolutionary and functional constraint on the position across multiple mammalian species. The variation occurs after the DNA binding domain and before the second nuclear localization signal of REST (Appendix A).

#### 3.4.2. Secondary Structural Changes in REST Due to p.C415S

Secondary structure analysis was performed in PSIPRED 4.0 [37,38]. The variation (Appendix A) caused substantial disruptions in multiple secondary structural features of the protein demonstrated by the black boxes in Figure 2B. This included (1) the complete abrogation of β strands at ^170^EEQ^172^, ^264^RKHLR^268^, ^434^EAD^436^, ^461^KNEKSVK^467^, and ^543^KKKK^546^, (2) the loss of helices at ^52^VA^53^, ^312^SS^313^, ^409^KS^410^, ^513^FS^514^, and ^522^KLEVD^526^, (3) disruptions in the composition of β strands within ^60^GSCCDYLVGEERQMAEL^76^ and ^171^EEQFVHHIRVH^181^ among other regions, and (4) disruptions in helices at ^105^GLEN^108^, ^501^EMDVH^505^, and ^493^RKSV^496^ among other regions. These secondary structural changes were predicted to cause disorder within a classic zinc finger (Znf) domain located between residues 360–439 based on a search against the InterPro [50] resource (Appendix A). This is expected to affect the overall function of REST.

#### 3.4.3. Conformational Changes in the Tertiary Structure Due to Variant REST

In-Silico modelling of the tertiary structure of the protein indicated that the variation resulted in structural changes in the protein. The variation results in a shift in the formation of disulphide bonds within the protein. It results in the loss of a disulphide bond between the cysteines at positions 415 and 1062 and the addition of a new bond between the cysteines at positions 337 and 363. It results in a change in the positions of the cysteines involved in all disulphide bonds, except for one, with bonds forming between new partners. It results in conformational changes in the protein when compared to the WT, whereby there is a difference in the predicted structure of the WT protein as compared to the MT. The tertiary structure of REST is indicated in Figure 2c,d, with the disulphide bond changes in disulphide bond formation indicated in Appendix A.

Furthermore, the mutation results in an extension of a disordered domain in the protein structure. This disordered domain extends in the last zinc finger in the mutant. In the wildtype, the disordered domain is, rather, adjacent to the zinc finger domain. This is indicated in Figure 2b.

### 3.5. In Vitro Functional Assay: Mutant REST Loses Exclusive Nuclear Localization and Ability to Repress Target

Using confocal microscopy (Figure 3a), the WT and MT REST proteins, tagged with GFP on the c-terminus, could be visualized in HEK-293 cells. As expected, cells transfected with an empty vector (expressing GFP only) showed strong and uniform GFP expression throughout the cells. The WT REST protein is located exclusively within the nucleus, as can be seen via co-localization of GFP-WT-REST with Hoechst, a DNA stain, as is expected of transcription factors. In contrast to this, the GFP-MT-REST displays a localization pattern similar to that of GFP-only, although to a seemingly lower intensity, indicating that the mutant REST protein losses exclusive nuclear shuttling/localization.

To compare the transcriptional repressive activity of the mutant REST with that of the WT, the ability of the proteins to repress a known REST target, *AF1q*, was assessed. WT or MT REST were transiently expressed in HEK-293 cells for 24 h, followed by qualitative PCR to measure levels of *AF1q* mRNA in these cells. This revealed that WT REST could competently repress *AF1q* transcription, by an average fold change of 0.51 (±0.07), while this transcriptional repression was lost in cells expressing the mutant REST protein, indicating that the mutation is of functional significance (Figure 3b).

Relative luciferase activity was also used to assess the ability of REST to repress AF1q. WT or MT REST were transiently expressed in HEK-293 cells for 36 h followed by luciferase assay. The assay revealed that WT REST could repress *AF1Q* transcription with a relative luciferase activity of 0.34 (±74.97), whereas transcriptional repression abated in cells expressing mutant REST protein. The difference in luciferase activity was however not significantly different (*p* 0.05).

## 4. Discussion

To our knowledge, this study is the first to investigate the association of HI with *REST* variants in individuals of African ancestry, and the second to demonstrate this gene’s association with ADNSHI globally. Thus, the data confirms *REST* as a novel HI gene. Moreover, the likely pathogenic variant reported is novel, and was not found in 52 unrelated sporadic cases of NSHI cases in a group of Black and Mixed ancestry South Africans. The data reinforce the genetic and locus heterogeneity nature of HI, and the urgent need of investigating diverse populations, particularly the understudied African populations, as the studies will help refining HI disease-gene curation.

*REST* encodes a transcription factor that represses the transcription of neuronal genes in non-neuronal cells [13,14,15]. The gene is associated with genetic stability and the lack of functional REST results in embryonic lethality during embryonic development [54,55]. REST is essential for neuronal development and a premature loss of REST results in the progenitor cells prematurely exiting the cell cycle [54]. Furthermore, REST is inactivated by alternative splicing, where the REST protein that contains the 4th exon is inactive [17].

In silico analyses show that the putative causative variation in exon 4 leads to changes in the protein structure (both secondary and tertiary), leading to re-arrangement of disulphide bonds, and protein folding. Transcription factors often work with other proteins within a complex to perform specific functions such as nuclear shuttling, recruitment of RNA polymerase and modelling of chromatin, and therefore a change in protein structure may lead to a disruption in any of these functions, which in turn impact target gene regulation [56]. *REST* is not only associated with HI, but it is also associated with a predisposition to Wilm’s Tumors [57] and has been implicated in colon cancer, small cell lung cancer, and neuroblastomas [58,59,60], and could suggest targeted long term follow up of the affected individuals.

In vitro assays showed the variants found in this family perturbs cellular localization of the protein. The fact that the mutant loses nuclear exclusivity may indicate a disruption in nuclear transportation. Although a nuclear localization signal (NLS) has been defined for REST (Appendix A), experiments by Shimojo (2006) showed that disruption of this domain did not affect nuclear shuttling, but instead disruption of the 5th zinc finger resulted in REST localizing to the cytoplasm [61]. The mutation may thus be responsible for disruptions to the nuclear localization machinery as modelling has indicated that the secondary structure and tertiary structure the DNA binding domain of REST (amino acids 159 to 412, see Appendix A) in the mutant is aberrated.

This change in REST structure may also be the cause of the loss of repression of the *AF1q* gene, a target of REST, when comparing the MT to the WT. This may be due to REST being unable to access its binding element, the RE1-like sequence in the *AF1q* promoter, which could be a result of the altered ability of the mutant REST to form complexes with partner proteins. REST has been defined as a master transcriptional regulator of neuron-specific genes via epigenetic remodeling, with the ability to recruit several partner proteins [62]. It is possible that the latter ability is impaired in the mutant REST protein. Future work should focus on elucidating the protein-protein and protein-DNA interactions of REST when considering the variation. The use of an HI in vivo model would be useful in further elucidating the pathogenic role of REST mutants in this disease. Causative variations in *REST* are rare, as only one other family has been reported [16], but Nakano et al. (2020) have shown that the HI phenotype in mice may be rescued if REST inactivation is prevented [63].

Although the variant identified in the present study is predicted to be deleterious (Appendix A) and affect the structure and function of the protein (Figure 2). The functional predication is further supported by an in vitro functional assay (Figure 3), more studies in other populations will likely inform and strengthen the HI disease gene-pair curation, globally.

## 5. Conclusions

We identified a monoallelic novel likely pathogenic variant in *REST* (OMIM: 600571). The variant NM_005612.4:c.1244GC co-segregated with non-syndromic autosomal dominant hearing impairment in an affected mother and son from South Africa. This study is the second report, worldwide, to describe the *REST*–HI gene-disease pair in humans, and thus confirms *REST* as a novel ADNSHI that should be included in targeted diagnostic gene panels. Our study emphasizes the urgent need of using WES to investigate hearing impairment in understudied African populations, to reveal the relevant valid gene variants to be investigated in clinical practice, and to enhance our understanding of hearing pathobiology, globally.

## Figures and Tables

**Figure 1 genes-12-01765-f001:**
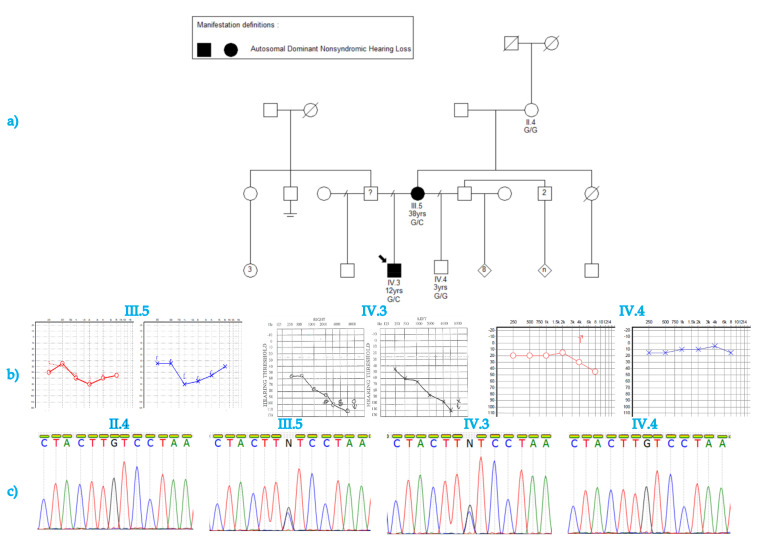
The pedigree of the family presenting with non-syndromic hearing impairment. (**a**) The pedigree suggesting a dominant inheritance model of hearing impairment (HI). The age and genotype are indicated under the ID for the four family members that were available for sequencing. (**b**) audiograms indicate severe hearing impairment in the affected mother and son and normal hearing in the unaffected child. The left ear is blue and the right ear is red in the audiograms for subject III.5 and IV.4. (**c**) The missense mutation segregates within the family with the affected mother and child being heterozygous for the c.1244GC. The unaffected grandmother and half-brother are homozygous wild-type.

**Figure 2 genes-12-01765-f002:**
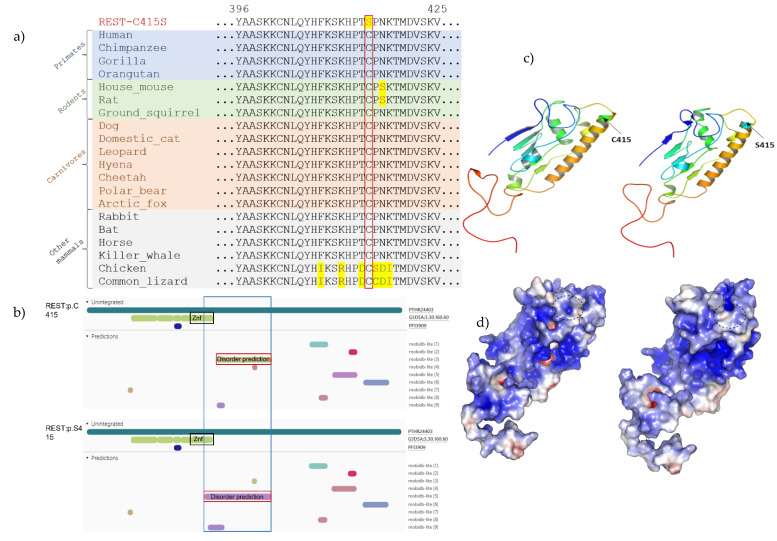
(**a**) The evolutionary conservation of the Cysteine amino acid at position 415 in REST. The position is indicated by the red box. (**b**) The domain structure of the WT and MT protein are shown. The mutant protein has an extended disordered domain when compared to the WT. (**c**) Protein modelling of REST comparing the WT (right) to the MT (left) in the ribbon form. The mutation results in changes in the β sheets and α helixes in the protein. This results in either the extension or the retraction of some the structures, as well as the formation and loss of other structures. (**d**) Protein modelling of REST comparing the WT (right) and the MT (left) in the space filling form indicating the consequences of the changes illustrated in (2c). The p.C415S variation results in a change in the tertiary structure of the protein such that it is slightly smaller than the WT protein. The MT protein also has several, previously exposed moieties, hidden.

**Figure 3 genes-12-01765-f003:**
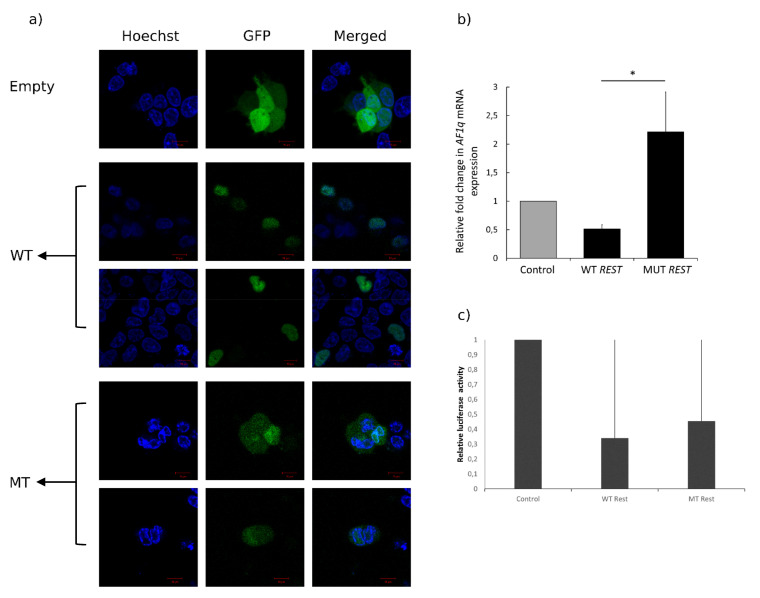
WT REST localizes to the nucleus, while MT REST displays localization patterns similar to GFP-only and loses repressive ability of target gene. (**a**) Micrographs of HEK-293 cells transfected with GFP-only (one representative image), GFP-tagged WT REST and GFP-tagged MT REST (two representative images each). The LHS row indicates Hoechst staining, the middle row indicates GFP, and the RHS row represents merged images. HEK-293 cells were transiently transfected with the respective constructs and visualized live after 24 h (spiked with Hoechst stain) under the confocal microscope (**b**) HEK-293 cells were transiently transfected with WT or MT REST expression constructs, and 24 h later the expression of AF1q mRNA was measured using qPCR and plotted relative to empty vector control. (**c**) HEK-293 cells were transiently transfected with WT and MT REST expression constructs and harvested and lysed 36 h later. Luciferase activity was measured with a luminometer and plotted relative to the empty vector control.

## Data Availability

The data presented in this study are available from the corresponding author upon reasonable request.

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
