# Peer review of "A Monoallelic Variant in REST Is Associated with Non-Syndromic Autosomal Dominant Hearing Impairment in a South African Family"

_genes, 2021, doi:10.3390/genes12111765_

Round 1

Reviewer 1 Report

In this article, authors find a probably damaging variant in REST locus in two related individuals with diagnosticated autosomal dominant hearing impairment. They use exome sequencing to elucidate candidate variants in this particular south african family and validate the functional implication of the selected variant through transfection in cell lines.

I recommend a major revision after some missing points that I adress next:

  1. First sentence of 2.6 paragraph is incomplete.
  2. Lack version of MODELLER and Pymol. It could be fine to add link to a visualization platform for a better understanding of the 3d model if possible. If it not possible due to journal limitations, a better image should be attached with higher resolution.
  3. Perhaps it may be interesting to test the possibility of a compound heterozygous inheritance. The finding of other candidate variants in the proband but not in the mother could lead to heterozygous variants inherited by the father modulating the severe phenotype of the proband. I understand probably father was unreachable, but possible compound heterozygous inheritance may be observable in variants present in IV.4, brother of the proband, in a trio analysis along with affected mother.
  4. It could be useful to add frequencies for the variant found for population datasets from gnomAD, such gnomAD African/African-American population, as it could be different from gnomAD global frequency. Besides that, authors chose a quite extreme rare MAF threshold for selection of rare variants (0.0005) when prevalence of the disease in south african population is noted as 0.005, quite higher. Perhaps a explanation about why that MAF threshold was used may be interesting. Also, complete localization for the variant should be added to the result paragraph (chr:start-endN>N), as well as reference genome used for the alignment.
  5. Figure 2 is difficult to understand. It should be added to supplementary. Quality should be improved. Figure 2A lacks a proper discussion about why choosing those species and why they decided to be sorted by alphabetical order instead of similarity percentage or something similar, where reader could appreciate better how conserved the region is (for instance, in mammals or primates). It is difficult to extract any relevant info from that part. Also, for conservation matters, also a measure of the constraint in REST may be added, using constraint metrics from gnomAD, for instance.
  6. Also, figure 2B relevancy may be not ideal for this article, as the ionic or hydrophobic status of the secondary structure of the entire protein is not relevant if the candidate aminoacid change does not affect it. This Figure 2B table could be more illustrative if, instead of covering a biochemical color legend for aminoacids not relevant for this study, author colors the functional domains of the gene or the disulphide bonds regions indicated in Figure 2D table for a better understanding of the possible functional effect in the architecture of the protein. Regarding Figure 2D, quality is low and it is almost impossible to read anything there.
  7. Also, in figure 2C, both protein models are a bit annoying to compare. I suggest adding a link to a online protein visualization to understand the differences in the 3D models if possible. If not possible, a zoom secondary image should be added for the region of interest. Also, calculation for bisulphide bonds were pointed in Figure 2D, so probably the respective disulphide bonds across the area may be pointed in the model for a better integration of the results.
  8. Figure 3 lacks a) and b) marks. Figure 3B qPCR plotting is not in the manuscript neither in the supplementary material.

Author Response

Reviewer 1

In this article, authors find a probably damaging variant in REST locus in two related individuals with diagnosticated autosomal dominant hearing impairment. They use exome sequencing to elucidate candidate variants in this particular South African family and validate the functional implication of the selected variant through transfection in cell lines.

 Authors’ response: Thanks very much for a far review of our manuscript

I recommend a major revision after some missing points that I address next:

  1. First sentence of 2.6 paragraph is incomplete.

Authors’ response: the missing word have ben added. Thanks!

  1. Lack version of MODELLER and Pymol. It could be fine to add link to a visualization platform for a better understanding of the 3d model if possible. If it not possible due to journal limitations, a better image should be attached with higher resolution.

 Authors’ response: We have now incorporated the model as follows:

The three-dimensional structure of the longest isoform of REST was used to generate protein models for the WT and MT REST. A homology model of the WT and MT REST was constructed using MODELLER (version 9.11)[41-44], based on the available crystal structure (6DU2 [45])as a template. PYMOL Viewer (version 2.4) [46, 47] was used for visualization of the structure and image processing. The disulphide bonds within the tertiary structure of REST were analyzed using DiANNA 1.1 Web Server [48-50]. Finally, the domains of REST were generated using InterPro [51].

  1. Perhaps it may be interesting to test the possibility of a compound heterozygous inheritance. The finding of other candidate variants in the proband but not in the mother could lead to heterozygous variants inherited by the father modulating the severe phenotype of the proband. I understand probably father was unreachable, but possible compound heterozygous inheritance may be observable in variants present in IV.4, brother of the proband, in a trio analysis along with affected mother.

 Authors’ response: We agree this is possible and have checked. We have revised the text as followed in the result section:

We have also checked the exome data for autosomal recessive variants in the affected proband. Exome data is available for the affected mother and proband, and the affected mother was used to phase variants in the proband. Homozygous or compound heterozygous variants with a MAF <0.005 were selected, in addition to known pathogenic and likely pathogenic variants reported in ClinVar (no MAF cut-off was used in that case). Exonic and splice site variants were evaluated, and we did not identify compound heterozygous nor homozygous variants in the proband in known hearing loss genes.

  1. It could be useful to add frequencies for the variant found for population datasets from gnomAD, such gnomAD African/African-American population, as it could be different from gnomAD global frequency. Besides that, authors chose a quite extreme rare MAF threshold for selection of rare variants (0.0005) when prevalence of the disease in south african population is noted as 0.005, quite higher. Perhaps a explanation about why that MAF threshold was used may be interesting. Also, complete localization for the variant should be added to the result paragraph (chr:start-endN>N), as well as reference genome used for the alignment.

Authors’ response:

  • A MAF cut-off of <0.0005 was used to filter for Autosomal Dominant variants only. Autosomal Recessive variants (both compound heterozygous and homozygous variants) were also evaluated with a MAF of <0.005. Known pathogenic and likely pathogenic variants reported in ClinVar were retained regardless of MAF. We have updated this in the manuscript:

“Rare variants for all populations in the genome aggregation database (gnomAD) with minor allele frequencies < 0.005 for autosomal recessive and <0.0005 for Autosomal Dominant inheritance were retained, in addition to known likely pathogenic and pathogenic variants in ClinVar, regardless of minor allele frequency.”

  • A complete localization for the variant is now provided: The reference genome used was GRCh37/hg19 and the variation is GRCh37:4:57796268:G:C
  • The variant was was absent from gnomAD, UK10K, Greater Middle East (GME) variome project databases, ClinVar as well as the Single Nucleotide Polymorphism Database (dbSNP).

  1. Figure 2 is difficult to understand. It should be added to supplementary. Quality should be improved. Figure 2A lacks a proper discussion about why choosing those species and why they decided to be sorted by alphabetical order instead of similarity percentage or something similar, where reader could appreciate better how conserved the region is (for instance, in mammals or primates). It is difficult to extract any relevant info from that part. Also, for conservation matters, also a measure of the constraint in REST may be added, using constraint metrics from gnomAD, for instance.

Authors’ response: We have now revised the figure 2 to provide more clarity and part of the figure have been moved to supplementary material (panels b, and d).

Specifically, the choice of species has been indicated (panel a), and constraint metric have been added.

  1. Also, figure 2B relevancy may be not ideal for this article, as the ionic or hydrophobic status of the secondary structure of the entire protein is not relevant if the candidate aminoacid change does not affect it. This Figure 2B table could be more illustrative if, instead of covering a biochemical color legend for aminoacids not relevant for this study, author colors the functional domains of the gene or the disulphide bonds regions indicated in Figure 2D table for a better understanding of the possible functional effect in the architecture of the protein. Regarding Figure 2D, quality is low and it is almost impossible to read anything there.

Authors’ response: The ribbon structure of the proteins has been added to show changes in the alpha helices and beta sheets due to the variant (new panel c).

  1. Also, in figure 2C, both protein models are a bit annoying to compare. I suggest adding a link to a online protein visualization to understand the differences in the 3D models if possible. If not possible, a zoom secondary image should be added for the region of interest. Also, calculation for bisulphide bonds were pointed in Figure 2D, so probably the respective disulphide bonds across the area may be pointed in the model for a better integration of the results.

Authors’ response: The space filling modelling has been moved to 2D to indicate the consequences of the changes of the alpha helices and beta sheets in the protein structure

  1. Figure 3 lacks a) and b) marks. Figure 3B qPCR plotting is not in the manuscript neither in the supplementary material.

Authors’ response: The qPCR plot has been included in the figure (3c).

In addition, a luciferase assay result plot has also been included (3d). HEK-293 cells were transiently transfected with WT and MT REST expression constructs and harvested and lysed 36 hours later. Luciferase activity was measured with a luminometer and plotted relative to the empty vector control.

Reviewer 2 Report

The manuscript entitled "A Monoallelic Variant in REST is Associated with Non-syndromic Autosomal Dominant Hearing Impairment in a South African Family" by Manyisa et al. is an interesting study on genetics of hearing loss in the South African population. The possible pathogenicity of the novel missense variant in REST gene is supported by numerous in silico prediction tools and also by in vitro functional assay. However, the authors should address following concerns: 

- Methods, page 2 and Table 2: "76 unrelated probands with sporadic NSHI of putative genetic origin comprising of both Black South African and South Africans of Mixed Ancestry (Table S2) were recruited, to investigate the frequencies of possible PLP variants identified." If the identified causative gene is responsible for sensorineural haering loss, it is unclear why the authors also included 2 subjects with pure conductive hearing loss, 7 subjects with mixed hearing loss and 22 subjects with unspecified hearing loss type. At least those with conductive and uspecified hearing loss should be removed or replaced by subjects with sensorineural hearing loss as they do not match the phenotype.  

- Results, page 5: There is a discrepancy between audiograms shown in the Figure 1 and the phenotypic description of the subjects. The authors state that "He (the proband) has progressive HI, air conduction thresholds had decreased in his 2019 audiogram when compared to his 2017 audiogram." This would actually mean that hearing in this subject has improved. It should be correctly written as "He has progressive HI, air conduction thresholds had increased in his 2019 audiogram when compared to his 2017 audiogram." Moreover, the sentence "He has severe HI in the right ear and profound HI in the left ear, both sensorineural." is also not true when looking at the audiogram (left ear seems to be better hearing than the right ear). Pure tone averages should be calculated for each ear. And subsequently, the authors state that "Her HI is sensorineural in her left ear and mixed in her right ear." This is not clear from the audiogram provided for subject III.5 and it should be explained. Similarly, the subject IV.4 is considered as unaffected, although the presented audiogram shows a high frequency hearing loss in one ear. In fact, it is difficult to distinguish which ear is left or right, because the use of audiogram colors seems to be opposite to what is used in the standard clinical practice (red for the right ear, blue for the left ear).   

- Results, page 5: "Based on the American College of Medical Genetics’ (ACMG) guidelines for the interpretation of sequence variants, the variant was classified as likely pathogenic." Regarding the variant classification, the use of hearing loss specific ACMG rules would be more appropriate (PMID: 30311386).

- The content of the table in the Figure 2d) is hardly visible. Images and figures need to be of higher quality.

Author Response

Reviewer 2

The manuscript entitled "A Monoallelic Variant in REST is Associated with Non-syndromic Autosomal Dominant Hearing Impairment in a South African Family" by Manyisa et al. is an interesting study on genetics of hearing loss in the South African population. The possible pathogenicity of the novel missense variant in REST gene is supported by numerous in silico prediction tools and also by in vitro functional assay. However, the authors should address following concerns: 

Authors’ response: Many thanks a fair summary of our manuscript and the positive comments.

- Methods, page 2 and Table 2: "76 unrelated probands with sporadic NSHI of putative genetic origin comprising of both Black South African and South Africans of Mixed Ancestry (Table S2) were recruited, to investigate the frequencies of possible PLP variants identified." If the identified causative gene is responsible for sensorineural haering loss, it is unclear why the authors also included 2 subjects with pure conductive hearing loss, 7 subjects with mixed hearing loss and 22 subjects with unspecified hearing loss type. At least those with conductive and uspecified hearing loss should be removed or replaced by subjects with sensorineural hearing loss as they do not match the phenotype.  

Authors’ response: The patients with solely conductive hearing impairment or unspecified hearing impairment have been removed from the analysis, and the descriptive statistics adjusted, accordingly.

- Results, page 5: There is a discrepancy between audiograms shown in the Figure 1 and the phenotypic description of the subjects. The authors state that "He (the proband) has progressive HI, air conduction thresholds had decreased in his 2019 audiogram when compared to his 2017 audiogram." This would actually mean that hearing in this subject has improved. It should be correctly written as "He has progressive HI, air conduction thresholds had increased in his 2019 audiogram when compared to his 2017 audiogram." Moreover, the sentence "He has severe HI in the right ear and profound HI in the left ear, both sensorineural." is also not true when looking at the audiogram (left ear seems to be better hearing than the right ear). Pure tone averages should be calculated for each ear. And subsequently, the authors state that "Her HI is sensorineural in her left ear and mixed in her right ear." This is not clear from the audiogram provided for subject III.5 and it should be explained. Similarly, the subject IV.4 is considered as unaffected, although the presented audiogram shows a high frequency hearing loss in one ear. In fact, it is difficult to distinguish which ear is left or right, because the use of audiogram colors seems to be opposite to what is used in the standard clinical practice (red for the right ear, blue for the left ear).

 Authors’ response: Pure tone averages for the proband and the mother have been included. The putative high frequency HI in subject IV.4 has been explained. And the colours of the audiograms have been included in the legend.

- Results, page 5: "Based on the American College of Medical Genetics’ (ACMG) guidelines for the interpretation of sequence variants, the variant was classified as likely pathogenic." Regarding the variant classification, the use of hearing loss specific ACMG rules would be more appropriate (PMID: 30311386).

Authors’ response: The guidelines used have been updated to the hearing loss specific ACMG rules.

- The content of the table in the Figure 2d) is hardly visible. Images and figures need to be of higher quality.

 Authors’ response: Figure 2 have been adjusted for clarity and, the table (panel d) has been moved to the supplementary data and the quality has been improved.

Round 2

Reviewer 1 Report

Authors have adressed most of my points.

Some minor issues:

Regarding hearing loss ACMG variant classification guidelines, it could be nice to add the criteria used at a new column in table S1 for each variant ( using the acronyms of each criteria for instance: PM1, BS2, etc...).

Figure S2 is strangely built. Black squares are displaced around the background overlapping some aminocids making it difficult to read. It could be improved to a more stylized image. Also, S2 footnote lack explanation about black square highligthing, as well as red and pink colors.

Author Response

Regarding hearing loss ACMG variant classification guidelines, it could be nice to add the criteria used at a new column in table S1 for each variant ( using the acronyms of each criteria for instance: PM1, BS2, etc...).

Authors' response. Thanks for your comments. The suggested column has been added to Table S1

Figure S2 is strangely built. Black squares are displaced around the background overlapping some aminocids making it difficult to read. It could be improved to a more stylized image. Also, S2 footnote lack explanation about black square highligthing, as well as red and pink colors

Authors' response. Thanks for noting this. We have provided an updated version of figure S2, with revised foot notes.

Reviewer 2 Report

Most of the previous concerns have been addressed by the authors. However, two more corrections have to be done:

Page 6: The last frequency in the two following sentences is not 200Hz, but 2000Hz.  "He has severe (pure tone average was evaluated as 71dB at 500, 1000 and 200Hz), sensorineural HI in both ears." "She presented with severe HI, in both ears (pure tone average was evaluated as 65dB in right ear and 67dB in left ear, at 500, 1000 and 200Hz), that was identified and diagnosed at 27 years of age."

Figure 1. The tags from above the audiograms have disappeared, so now it is not clear which audiogram belongs to which subject from the pedigree. The same is true for fig. 1c. Moreover the two audiograms (the right one and the left one) have opposite positions of charts for left and right ear. Left ear (blue) should be on the right side and right ear (red) should be on the left side, because you look at the audiogram as if you looked at the patient sitting in front of you face to face. Look at the audiogram in the center, which is correct.

Author Response

Most of the previous concerns have been addressed by the authors. However, two more corrections have to be done:

Authors' response. Many thanks for your fine editing.

Page 6: The last frequency in the two following sentences is not 200Hz, but 2000Hz.  "He has severe (pure tone average was evaluated as 71dB at 500, 1000 and 200Hz), sensorineural HI in both ears." "She presented with severe HI, in both ears (pure tone average was evaluated as 65dB in right ear and 67dB in left ear, at 500, 1000 and 200Hz), that was identified and diagnosed at 27 years of age."

Authors' response. The text I now revised, accordingly. Thanks!

Figure 1. The tags from above the audiograms have disappeared, so now it is not clear which audiogram belongs to which subject from the pedigree. The same is true for fig. 1c. Moreover the two audiograms (the right one and the left one) have opposite positions of charts for left and right ear. Left ear (blue) should be on the right side and right ear (red) should be on the left side, because you look at the audiogram as if you looked at the patient sitting in front of you face to face. Look at the audiogram in the center, which is correct

Authors' response. Figure 1 has been revised, accordingly. Thanks.